# Genomic Screening Reveals That the Endangered *Eucalyptus paludicola* (Myrtaceae) Is a Hybrid

**Kor-jent van Dijk** [1] , **Michelle Waycott** [1,2,*] , **Joe Quarmby** [3,4], **Doug Bickerton** [3],
**Andrew H. Thornhill** [1,2] , **Hugh Cross** [1,5] **and Edward Biffin** [2]

1 School of Biological Sciences, University of Adelaide, Adelaide, SA 5005, Australia;
korjent.vandijk@adelaide.edu.au (K.-j.v.D.); andrew.thornhill@adelaide.edu.au (A.H.T.);
hugh.cross@otago.ac.nz (H.C.)
2 Department for Environment and Water, Botanic Gardens and State Herbarium, State Herbarium of
South Australia, Adelaide, SA 5001, Australia; edward.biffin@adelaide.edu.au
3 Department for Environment and Water, Government of South Australia, Adelaide, SA 5001, Australia;
jquarmby@tasland.org.au (J.Q.); doug.bickerton@sa.gov.au (D.B.)
4 Tasmanian Land Conservancy, Lower Sandy Bay, TAS 7005, Australia
5 Department of Anatomy, University of Otago, Dunedin 9054, New Zealand
* Correspondence: michelle.waycott@adelaide.edu.au

**Abstract:** A hybrid origin for a conservation listed taxon will influence its status and management options. Here, we investigate the genetic origins of a nationally endangered listed taxon—*Eucalyptus paludicola*—a tree that is restricted to the Fleurieu Peninsula and Kangaroo Island of South Australia. Since its description in 1995, there have been suggestions that this taxon may potentially be a stable hybrid species. Using a high throughput sequencing approach, we developed a panel of polymorphic loci that were screened across *E. paludicola* and its putative parental species *E. cosmophylla* and *E. ovata*. Bayesian clustering of the genotype data identified separate groups comprising *E. ovata* and *E. cosmophylla* while *E. paludicola* individuals were admixed between these two, consistent with a hybrid origin. Hybrid class assignment tests indicate that the majority of *E. paludicola* individuals (~70%) are $F_1$ hybrids with a low incidence of backcrossing. Most of the post-$F_1$ hybrids were associated with revegetation sites suggesting they may be maladapted and rarely reach maturity under natural conditions. These data support the hypothesis that *E. paludicola* is a transient hybrid entity rather than a distinct hybrid species. We briefly discuss the conservation implications of our findings.

**Keywords:** conservation; natural hybridisation; *Eucalyptus*; high throughput sequencing; NGS; South Australia

## 1. Introduction

Natural interspecific hybridisation is a common phenomenon in plants and is an important evolutionary process. The outcomes of hybridisation can be diverse, maintaining, reducing or increasing evolutionary divergence among taxa [1]. Hybridisation between well-differentiated species can lead to the origin of new species involving a change in base chromosome number (i.e., allopolyploid hybrid speciation) or without such a change (homoploid hybrid speciation). In the case of the former, a change in ploidy can lead to the spontaneous development of reproductive isolation, while in contrast successful homoploid hybrid species must overcome significant ecological, genetic and demographic obstacles and are thus considered rare [2]. Homoploid hybrid speciation has been suggested in a number of cases, although the majority of these have demonstrated a hybrid origin of taxa but lack

conclusive evidence for hybrid speciation [1]. Key additional criteria include evidence of reproductive isolation of hybrid lineages from the parental species and evidence that reproductive isolation arose as a consequence of hybridisation [2].

The genus *Eucalyptus L'Hér.* includes approximately 650 species of mostly trees that are endemic to Australia (with a few exceptions) and dominate the forests and woodlands of that continent. Taxonomists classify eucalypt species, including members of the three genera: *Angophora Cav.*, *Corymbia K.D.Hill & L.A.S.Johnson,* and *Eucalyptus*, into various subspecies, sections and series based on morphology and assumed relatedness (see Nicolle 2019 [3] for the most recent classification). Eucalypts are the worlds' most widely grown plantation hardwoods and there has been considerable interest in genetic improvement through manipulated hybridisation [4]. Natural interspecific hybridisation has also been widely reported within the group (e.g., [5–9]) and the propensity for hybridisation and its outcomes are strongly predicted by the degree of evolutionary divergence among species [5,10–13]. Natural hybridisation between species from different genera or subgenera is not believed to occur (but see [7] for a possible exception) and is relatively uncommon among sections within subgenera [5,13]. There are several well-documented cases of natural hybridisation amongst closely related eucalypts although according to Griffin et al. [5] only 15% of cases, where hybridisation was likely (intrasectional relatives with proximal distributions), resulted in hybrid formation. Known prezygotic barriers in eucalypts include pollen tube arrest, the frequency of which increases with evolutionary distance between parents [14] while species-specific variation in flower size presents a structural barrier to gene flow [15]. Postzygotic barriers may also contribute to reproductive isolation. In a recent study, there was strong evidence for outbreeding depression (negative epistasis) amongst two closely related *Eucalyptus* species [16], which is predicted to increase with evolutionary divergence because of "snowballing" epistatic interactions [17].

*Eucalyptus paludicola* D.Nicolle was described by Nicolle 1995 [18] and in light of its rarity and exposure to ongoing threatening processes [19] has been listed as endangered under the Environment Protection and Biodiversity Conservation Act of 1999 (Commonwealth of Australia) and the National Parks and Wildlife Act 1972 (State of South Australia). The known distribution includes approximately 34 populations with an estimated 720–750 individuals in total [20]. Its range is limited to the Fleurieu Peninsula and Kangaroo Island of South Australia, where it occurs on seasonal swampy sites, often in highly modified landscapes. *E. paludicola* is commonly coassociated with *Eucalyptus cosmophylla F. Muell.* and *E. ovata Labill.* and is in some respects morphologically and ecologically [18] as well as genetically (based on amplified fragment length polymorphisms [21]; and DART markers [22]) intermediate between these species, suggesting the likelihood of a hybrid origin. The relatively constant morphology of *E. paludicola* throughout its distributional range and evidence that the progeny breed true in an arboretum has been argued as supporting specific recognition [18,23]. While existing molecular evidence is consistent with a hybrid origin, a hypothesis of hybrid speciation is difficult to distinguish from alternatives [2], for instance, that *E. paludicola* individuals represent transient hybrids lacking reproductive isolation from the parents. Given that there are presently no known instances of hybrid speciation among eucalypts, a resolution of this issue may have significant bearing on our understanding of hybridisation and speciation within this ecologically and economically important lineage. Resolving the origins of *E. paludicola* is also important in clarifying its taxonomic and conservation status. In Australia, if a species is listed under the Federal Environment Protection and Biodiversity Conservation Act 1999 (EPBC Act), then it must be protected. Further, species that have arisen via hybridisation may be afforded full conservation protection while transient hybrid (i.e., lacking reproductive isolation from the parents and hybridisation not directly leading to the formation of a new taxon) entities are not presently recognised under relevant Commonwealth and State legislation. The EPBC Act does not, however, require a test to identify if that species might be a transient hybrid or hybrid species, meaning some currently listed taxa may be invalid.

To resolve the origins of *E. paludicola* we used a high throughput sequencing (HTS) approach to assess genetic diversity. Specifically, we leveraged the *E. grandis* genome sequence [24] and

restriction-site based HTS to develop a panel of molecular markers. We used multiplex PCRs to screen across representatives of *E. paludicola* along with *E. ovata* and *E. cosmophylla* and applied Bayesian statistical approaches to hybrid assessment from these data. If *E. paludicola* is a hybrid species it will be indicated by factors including unique genetic diversity and coherence across its geographical range. The alternative hypothesis, that this taxon is a transient hybrid, would be supported by a predominance of early hybrid generations (e.g., $F_1$ individuals) consistent with the absence of reproductive isolation.

## 2. Materials and Methods

### 2.1. Study Species

We sampled individuals of *E. paludicola, E. cosmophylla* and *E. ovata* across the region of geographic sympatry that includes the southern Fleurieu Peninsula and Kangaroo Island in the state of South Australia. Both *E. paludicola* and *E. cosmophylla* are endemic to this region, while *E. ovata* is widely distributed in temperate south-eastern Australia, with a disjunct range extending to our study region. All three species are placed within *Eucalyptus* subg. *Symphyomyrtus,* and *E. paludicola* and *E. cosmophylla* are considered to be close relatives (Section *Incognitae*; [18]) while *E. ovata* is placed within Section *Maidenaria. E. paludicola* and *E. cosmophylla* can be readily distinguished by their habit (taller and more upright, versus a shrubby to mallee habit, respectively) and inflorescence structure (usually seven flowered in *E. paludicola* versus consistently three flowered in *E. cosmophylla*), while *E. paludicola* has thinner leaves, longer peduncles and pedicels and smaller flowers and fruits. *E. ovata* has a tree habit, a consistently seven flowered inflorescence and has smaller flowers and fruit than the other two species. Both *E. paludicola* and *E. ovata* are associated with seasonally swampy sites, while *E. cosmophylla* grows on a range of soil types from infertile sands to poorly drained gravelly clays [18,23].

### 2.2. Sample Collection and DNA Extraction

Leaf material was collected from individuals of each species on both Fleurieu Peninsula and Kangaroo Island and dried using silica-gel. Genomic DNA was subsequently extracted from the leaf material using a commercial service (Australian Genome Research Facility, Adelaide). Sampling details are included in Table 1 (additional details in supplementary Table S1). Our sampling included two sites that were planted with *E. paludicola* as part of a restoration project (*Eucalyptus paludicola* recovery program, Department for Environment and Water).

### 2.3. Marker Development and Sequencing

We used methods described by Cross et al. [25] to generate a sequencing library for representative samples of *E. cosmophylla, E. ovata* and *E. paludicola.* The library was then sequenced using an Ion Torrent Personal Genome Machine (Ion PGM, Life Technologies, Carlsbad, California, USA) with 200 bp sequencing chemistry and a 318 v.1 sequencing chip. The raw sequences were imported into CLC Genomics Workbench 9 (Qiagen, Venlo, The Netherlands) for demultiplexing, adapter and quality trimming (ambiguous trim = 2, quality limit = 0.05, minimum read length = 50). The trimmed reads were then de novo assembled (indel and mismatch cost = 2; similarity fraction = 0.8, length fraction = 0.5) and we extracted contigs with coverage greater than 100×. The resulting contigs were compared to the *E. grandis* v2.0 genome sequence [24] using *blast-n* (default parameters in CLC) to identify regions that, from our sequencing library, had a single blast hit and could be considered as potential single copy regions in our target eucalypt species. From these, we targeted regions that had one or more SNPs in our assemblies and had suitable priming sites (based upon the *E. grandis* genome sequence; supplementary Table S2) to amplify a product of not more than 150 bp (excluding primer, adapter and barcode sequences), which is the approximate limit of the Ion Torrent PGM 200 bp sequencing chemistry. Primers were designed in Geneious v7 (Biomatters, Auckland, New Zealand) [26] using the Primer 3 [27] plug-in.

**Table 1.** Taxa and locations of tested samples.

| Taxon | Location | *n* | Latitude | Longitude |
|---|---|---|---|---|
| *E. cosmophylla* | Three Chain Rd (KI) | 6 | −35.829 | 137.704 |
| *E. cosmophylla* | Crafers West (FL) | 2 | −34.990 | 138.681 |
| *E. cosmophylla* | Kyeema (FL) | 4 | −35.270 | 138.674 |
| *E. cosmophylla* | Mt Billy Cp (FL) | 2 | −35.448 | 138.600 |
| *E. cosmophylla* | Burnfoot (FL) | 4 | −35.400 | 138.557 |
| *E. ovata* | Burnfoot (FL) | 3 | −35.400 | 138.557 |
| *E. ovata* | Cleland Gully Rd (FL) | 2 | −35.368 | 138.638 |
| *E. ovata* | Stipiturus CP (FL) | 1 | −35.371 | 138.551 |
| Undetermined | Stipiturus CP (FL) | 1 | −35.371 | 138.551 |
| *E. paludicola* | Stipiturus CP (FL) | 1 | −35.371 | 138.551 |
| *E. paludicola* | Kelly Hill Caves CP (KI) | 15 | −35.997 | 136.873 |
| *E. paludicola* | Short's property, original tree (KI) | 1 | −35.739 | 137.029 |
| *E. paludicola* | Short's property, revegetation (KI) | 9 | −35.739 | 137.028 |
| *E. paludicola* | Edwards Lagoon (KI) | 4 | −35.808 | 137.038 |
| *E. paludicola* | O'Donnell property (KI) | 5 | −35.963 | 136.999 |
| *E. paludicola* | Rocky River, natural (KI) | 4 | −35.946 | 136.753 |
| *E. paludicola* | Rocky River revegetation (KI) | 9 | −35.947 | 136.741 |
| *E. paludicola* | Nangkita (FL) | 11 | −35.345 | 138.711 |
| *E. paludicola* | Burnfoot (FL) | 8 | −35.400 | 138.557 |
| *E. paludicola* | Gold Diggings Swamp (FL) | 2 | −35.591 | 138.372 |
| *E. paludicola* | Hindmarsh Valley (FL) | 5 | −35.402 | 138.518 |
| *E. paludicola* | Range Rd (FL) | 5 | −35.567 | 138.469 |
| *E. paludicola* | Parawa (FL) | 2 | −35.591 | 138.372 |
| *E. paludicola* | Mosquito Hill Rd (FL) | 19 | −35.446 | 138.646 |
| *E. paludicola* | Kokoda Rd (FL) | 10 | −35.407 | 138.688 |
| *E. paludicola* | Cox Scrub CP (FL) | 9 | −35.331 | 138.747 |
| *E. paludicola* | Proctor Road (FL) | 7 | −35.326 | 138.621 |

*n* = number of samples; KI, Kangaroo Island; FL, Fleurieu Peninsula.

We used a fusion PCR approach to generate an amplicon sequencing library for each individual [28]. Briefly, in a first step, we amplified in multiplex two panels of primers (24 primer-pairs each at 2 μM per primer in mix, supplementary Table S2), In a second step, we added a sample specific barcode by fusion PCR using the universal adapters added to the locus specific PCR primers as priming sites.

In more detail; for the first round of multiplex PCR we used the Multiplex PCR kit of Qiagen (Venlo, Netherlands) using the suggested manufacturers cycling conditions, with annealing temperature at 60 °C and amplifying 20 cycles to avoid overamplification. The reactions were done in 10 μL reactions adding 1 μL of primer mix (2 μM equimolar primer concentrations, supplementary Table S2) and 1.5 μL undiluted DNA template. Each primer consisted of the original sequence obtained from the *E. grandis* genome and an adapter sequence extension on the 5′ end to allow fusion PCR (*Eco*RI adaptor on forward and *Mse*I adapter on reverse, supplementary Table S4 in [25]). For the second round of PCR the same chemistry was used as in the first adding 1 μL EcoRI + CA and 1 μL *Mse*I + C fusion primers with internal barcodes (following supplementary Table S4 and Selective amplification in [25]) and 1μL of template (amplicons of first PCR). The reactions were done in 10 μL reaction with an annealing temperature of 60 °C and 15 cycles of amplification.

The resulting individual libraries were pooled, purified with AMPure XP (Agencourt, Beckman Coulter Inc., Brea, CA, USA) and then quantified using a 2200 TapeStation (Agilent, Santa Clara, CA, USA) with a high-sensitivity ScreenTape. The final library was diluted to 9.0 pMol and sequenced on an Ion PGM 318 v.1 chip. A total of 5 libraries were prepared to finalise the study.

*2.4. Data Processing*

The raw sequence data were imported into CLC Genomics Workbench 9 for demultiplexing, adapter and quality trimming (ambiguous trim = 2, quality limit = 0.05, minimum read length = 50). The resulting reads for each individual were then mapped to the reference sequences (indel and

mismatch score = 2, similarity fraction = 1.0, length fraction = 0.5). With these mappings as input we used the fixed ploidy variant caller in CLC to identify SNPs (ploidy = 2, required variant probability = 0.95, minimum coverage = 10, minimum frequency = 0.2, filter homopolymer regions with minimum length = 3). Read mappings were then visually inspected and haplotypes were manually determined. For the final data we excluded loci that showed evidence of paralogy (more than 2 haplotypes per individual), amplified poorly (coverage <10 for >40% of individuals) or were invariant (minor allele frequencies <0.2). For all downstream data analyses, we treated each haplotype recovered as an allele for that locus.

*2.5. Data Analyses*

Summary statistics for species level genetic parameters including observed number of alleles (A) and allele frequencies, unbiased gene diversity ($H_E$), observed heterozygosity ($H_O$) and the fixation index ($F_{IS}$) were calculated using the software GenAlEx version 6.5 [29]. For these analyses, individuals were assigned to species groups based upon their morphological identification with the exception of those which, based upon the STRUCTURE results with K = 2, had an assignment probability to a cluster of <0.9.

Bayesian clustering analysis implemented in the program STRUCTURE 2.3.4 [30] was used to identify the most likely number of genetic clusters (K) among the samples and to assign individuals to clusters (input data in supplementary Table S3). Using this approach, individual genotypes can assign to a single cluster or can have mixed assignment when ancestry is shared in more than one parental group due to hybridisation (i.e., admixture). For these analyses, the admixture model with correlated allele frequencies with no priors of individual identification (i.e., morphological species assignment) were used. Ten independent runs at K values one to eight were run with MCMC simulations having 900,000 steps following a burn-in period of 100,000. The delta-K statistic [31] as implemented in STRUCTURE HARVESTER [32] was used to determine the most likely number of clusters (K) in the data. We assessed the average proportion of membership ($Q_i$) of samples to the inferred cluster and the individual membership proportion $Q_i$ of each sample to the K clusters.

The methods implemented in NEWHYBRIDS 1.1 [33] were used to assess the evidence of hybrid ancestry for individual samples (input data in supplementary Table S3). NEWHYBRIDS uses MCMC simulations to provide a posterior probability of an individual assignment to predefined genealogical classes. We ran these analyses assuming 2 generations of hybridisation resulting in six genealogical classes: one class for each parental type as well as first generation ($F_1$), second generation ($F_2$) and 2 backcross ($F_1 \times$ parent 1; $F_1 \times$ parent 2) hybrid classes. All analyses were run without prior information regarding individual membership. Analyses were run over 200,000 steps following 50,000 burn-in using "Jeffery's like priors" for mixing proportions and allele frequencies. Three replicate analyses were performed to assess consistency across runs. We used a posterior probability of 0.9 as a threshold for assigning an individual to a specific genealogical class.

Following Nielsen et al. [34] we conducted simulations for our empirical data in order to assess the power of these markers to distinguish among genealogical classes and assess the range of q values expected for admixed individuals. HYBRIDLAB [34] was used to simulate pure parental genotypes using real data from reference individuals that could be unambiguously assigned (individual assignment probability >0.95) to a genetic cluster in our STRUCTURE analyses (above) with K = 2. We simulated 200 genotypes for each parental type that were then used to successively generate $F_1$, $F_2$ and two backcross genotype classes comprising 200 simulated multilocus genotypes per genealogical class. The simulated data where then analysed using STRUCTURE, with K set to 2, in order to assess the efficiency and accuracy of these analyses to identify admixed genotypes. Similarly, simulated genotypes were analysed in NEWHYBRIDS to assess the efficiency with which this approach could allocate simulated individuals to the correct genotype class. For both sets of analyses, settings are as described above for the real genotype data. Finally, a Principal Coordinates Analysis was performed in

GenAlEx to visualise the separation between species and individuals relative to simulated data using 50 simulated genotypes per genotype class.

## 3. Results

### 3.1. Molecular Data

A suite of 30 gene regions that were consistently amplifying with high coverage were used from a set of 48 primer pairs developed. These 30 gene regions generated adequate sequence data, containing multiple SNP loci, for downstream analyses. Of the eighteen loci deemed not adequate, 13 amplified poorly or not at all, 4 were invariant, and for one locus more than 2 alleles were apparent in some individuals suggesting paralogy. From the 30 useable marker gene regions, the proportion of missing data per locus averaged <4%, with a single locus having a maximum of c. 40% while the remaining loci generally fell below 10% of missing values. Missing values per individual were on average <10% (range 0–70%). A final dataset of 151 individuals was used for further analyses (supplementary Table S1).

The number of alleles per locus ranged from 2 to a total of 21 alleles (supplementary Table S1). The average number of alleles across the 30 loci was highest for *E. paludicola* (7.17), followed by *E. cosmophylla* (3.1) and *E. ovata* (2.3). This presumably reflects differences in sample size included in this study across taxa, as well as a broader geographic sampling of *E. paludicola.* However, *E. paludicola* was heterozygous at all loci (observed heterozygosity, 0.017–0.968, average = 0.664) while *E. cosmophylla* and *E. ovata* were monomorphic at 5 and 9 loci, respectively, and had average observed heterozygosity of 0.319 (range, 0–0.833) and 0.301 (range, 0–1), respectively. Both *E. cosmophylla* and *E. ovata* showed complete segregation at 8 loci (c. 30%), while there were no markers that were diagnostic for *E. paludicola.* *E. cosmophylla* and *E. ovata* were strongly diverged with an $F_{ST}$ of 0.453, in line with intersectional comparisons within subg. *Symphyomyrtus* [35], while $F_{ST}$ values for *E. paludicola* were 0.131 (*E. cosmophylla*) and 0.167 (*E. ovata*).

### 3.2. Admixture Analyses and Hybrid Class Assignment

The delta-K statistic for the admixture analyses performed using STRUCTURE was clearly optimal for K = 2 (ΔK = 855.9) with the next best grouping being four clusters (ΔK = 10.9) (Figure S2 in supplementary Table S4). When 2 groupings were assumed, individuals morphologically identified as *E. cosmophylla* or *E. ovata* were each unambiguously assigned to a single distinct cluster. Individuals identified as *E. paludicola* were partially assigned to both clusters with individual membership proportions $Q_i$ < 0.9 (Figure 1). Significantly, when 3 clusters were assumed, the *E. paludicola* samples did not form a discrete group, as would be expected under the hypothesis that it is a distinct taxon (data not shown). Using simulated data with two parental and 4 hybrid categories, the average membership proportions for parental genotypes were >0.98 to a single K cluster while average assignment probabilities for each of the simulated hybrid classes was <0.91 (maximum individual assignment probability = 0.9 for *E. ovata* backcross) (supplementary Table S5). The $Q_i$ values of the actual genotype data for *E. paludicola* fell within the range of values found for simulated hybrids and below the values of simulated parental genotypes (Figure 1) and it is reasonable to conclude that most *E. paludicola* individuals can be classified as admixed.

The analyses of the simulated genotype data using NEWHYBRIDS assigned virtually all samples to their expected genealogical class with posterior probabilities exceeding 0.95 (supplementary Table S5) and in the few cases with lower confidence there were no misclassified individuals (i.e., probabilities were apportioned among classes). This suggests that the empirical genotype data contain adequate signal to correctly assign individuals to parental and various hybrid categories. Analyses of the empirical data using NEWHYBRIDS (Figure 2) closely reflect the results of the admixture analyses above (Figure 1) and in particular, individuals identified as *E. cosmophylla* were unambiguously assigned to a pure parental class (posterior probability >0.99), as were all but one *E. ovata* sample. The admixed

individuals (*E. paludicola* and one *E. ovata* individual) identified by STRUCTURE were assigned to a range of hybrid classes with most individuals being placed in a single category. Of the 126 *E. paludicola* individuals included in these analyses, 90 (*c.* 71%) were unambiguously assigned (posterior assignment probability >0.9, mean assignment probability >0.99) to an $F_1$ class and 15 individuals (*c.* 12%) were identified as $F_2$ hybrids. The three individuals (c. 2.5%) identified as backcrossed to *E. ovata* were associated with revegetation sites, as were c. 75% of the $F_2$ individuals. Eighteen individuals (*c.* 14%) could not be confidently placed within a single genotypic class (assignment probabilities <0.9) but were fractioned between pure *E. cosmophylla* (2 individuals), $F_1$, $F_2$ and backcross classes (Figure 2). Assignment probabilities for each genealogical class were summed across all individuals to give an expected number of *E. paludicola* plants in each category (supplementary Table S3 Figure S3). These analyses indicate the predominance of $F_1$ (~76%) and $F_2$ genotypes (~17%) and a relatively low incidence of backcrossing (~3.5%), particularly in the direction of *E. cosmophylla* (~1.9%). Figure 3 plots the first two principal coordinates of 2 analyses (data in supplementary Table S1), one with the real data (all eucalypts used in study) and the 50 simulated genotypes for each genealogical class. The PCoA analysis showed a clear separation of *E. ovata*, *E. cosomophylla* and *E. paludicola*, which is placed in the intermediate position between them. For the simulated genotype data, the $F_1$ and $F_2$ classes cluster with *E. paludicola*, while the backcross classes are intermediate between *E. paludicola* and each parental taxon. More than 30% of the variation is explained by these two coordinates, the majority being on the horizontal axis (Figure 3). This supports the most important signal in the data being the relative placement of *E. ovata*, *E. cosomophylla* and *E. paludicola* and aligns the latter with the $F_1$ hybrids as determined by simulations.

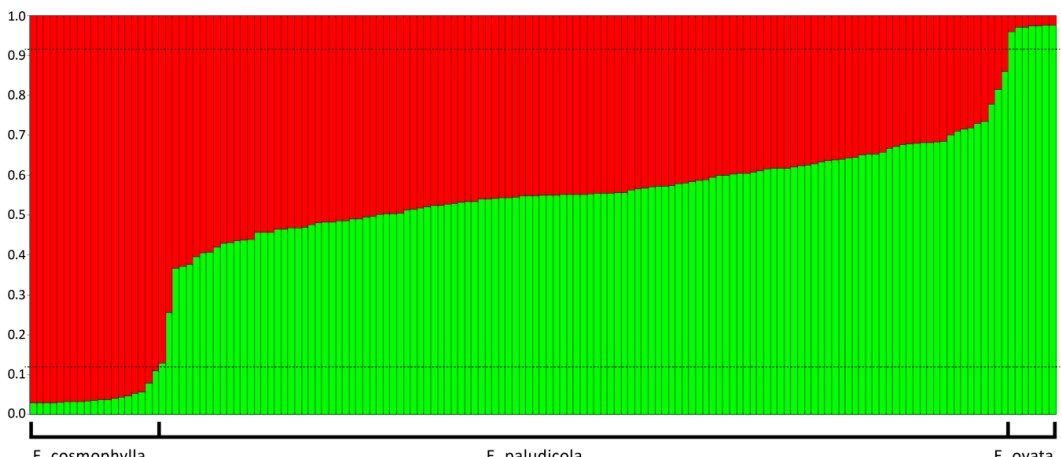

**Figure 1.** Proportion of ancestry for 151 *Eucalyptus* individuals with K = 2, inferred using STRUCTURE. Red and the green clusters correspond to *E. cosmophylla* and *E. ovata,* respectively, while *E. paludicola* shows admixture. The horizontal dashed lines indicate the range of *Q* values for simulated hybrid individuals (see text for details).

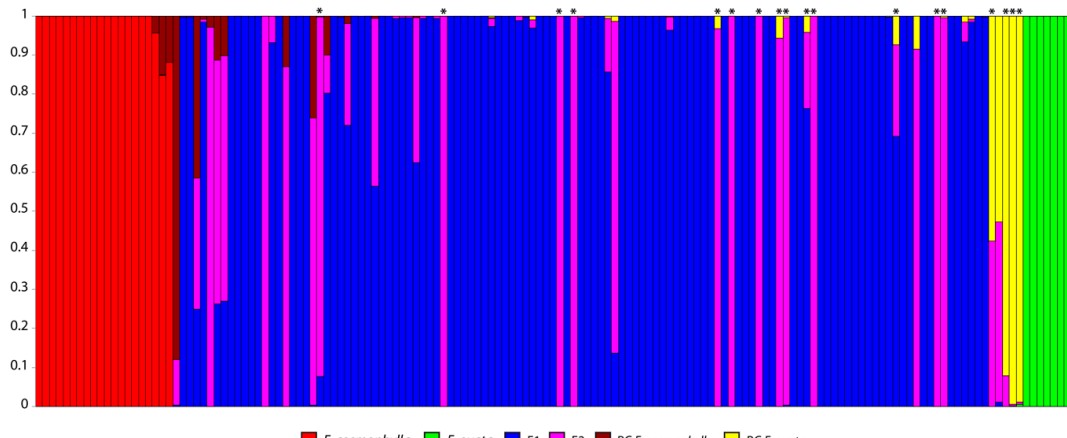

**Figure 2.** The proportion of individual assignment to 6 predefined genealogical classes for 151 *Eucalyptus* individuals, inferred using NEWHYBRIDS and 30 loci. Red and the green clusters correspond to "pure" *E. cosmophylla* and *E. ovata*, respectively, while *E. paludicola* is represented by a range of hybrid classes. The asterisks (*) correspond to *E. paludicola* individuals sampled from revegetation sites. BC in the figure stands for "back cross".

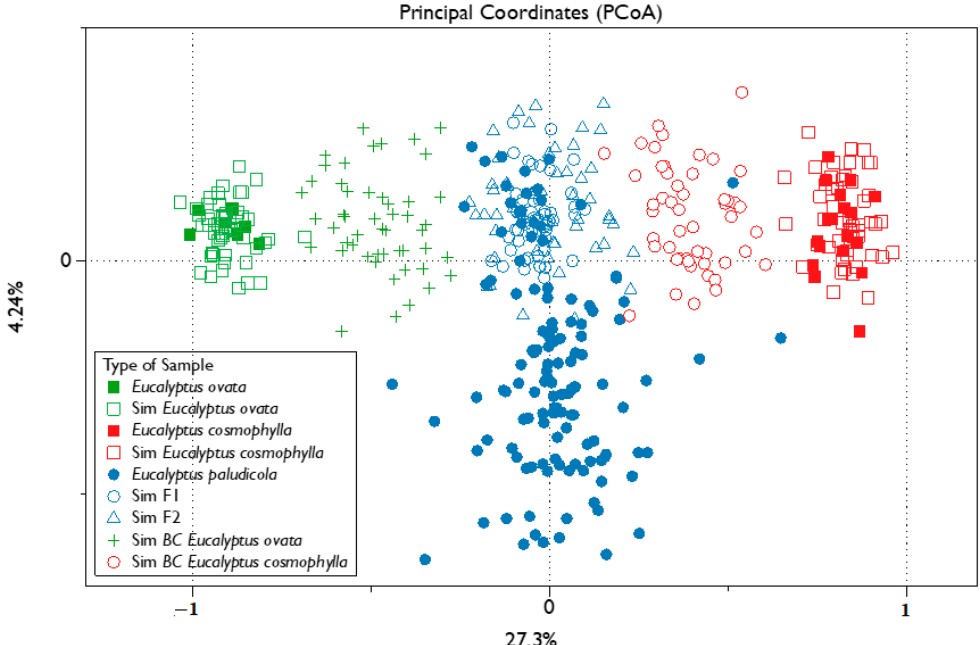

**Figure 3.** Principal Coordinates Analysis for real and simulated genetic data for *Eucalyptus paludicola*, *E. ovata*, *E cosmophylla*. First ($F_1$) and second ($F_2$) generation, back cross with *E. ovata* and back cross with *E. cosmophylla* were all simulated in HYBRIDLAB to depict the positioning of *E. paludicola*.

## 4. Discussion

The results of our study clearly indicate that *E. paludicola* is a hybrid taxon derived from *E. cosmophylla* and *E. ovata*, as has been previously suspected based upon morphological [18,23] and molecular evidence [21,22]. Despite the relatively high frequency of hybridisation among eucalypts, the formation of this hybrid would be unexpected given a relatively deep divergence between the progenitor species—*E. cosmophylla* and *E. ovata* are placed within different taxonomic sections of subg. *Symphyomyrtus*, which among eucalypts, strongly predicts the potential likelihood of successful hybrid formation [5,12,17]. Cases of natural intersectional hybridisation are relatively uncommon, and for example, Griffin et al. [5] reported intersectional hybrids within subg. *Symphyomyrtus* at a frequency of <5% amongst geographically proximal species. *E. paludicola* thus represents a rare case of natural

intersectional hybridisation. A recent study based upon crossing experiments between *E. globulus* and congeners spanning a broad range of evolutionary distances suggests that complete reproductive isolation among eucalypts may take in excess of 20 million years to develop [17] (divergence time estimates based upon [36], but see Thornhill et al. [37], who infer significantly younger divergence times for eucalypt lineages based on a more densely sampled phylogeny). According to Crisp et al. [36] the divergence of section *Maidenaria* (Clade I of Steane et al. [38]; *E. ovata*) and the closest relations of section *Incognitae* (*E. cosmophylla*) that were included in their study (sect. *Exsertaria* and *Bisectae,* Clade II of Steane et al. [38]) occurred at between 5–15 million years before present, providing an approximate timescale for the divergence of *E. cosmophylla* and *E. ovata.*

These results indicate that the proposal for a possible instance of *Eucalyptus paludicola* as a hybrid speciation, so far unknown among eucalypts, is unsupported. To confirm this hypothesis, it would need to be supported by evidence that hybridisation has led to the development of reproductive isolation [2]. In addition to a probable hybrid origin, circumstantial evidence that *E. paludicola* is reproductively isolated from the parental taxa [22] has included limited variation in adult and seedling morphology throughout its geographical range, arguing against introgression and the observation that progeny planted in an arboretum show levels of morphological variation within the range of that taxon [18,22]. However, our findings suggest a contrasting interpretation of the status and origins of *E. paludicola*. In particular, the predominance of $F_1$ hybrids among *E. plaudicola* individuals included in this study (Figure 2) is consistent with a relatively uniform morphology that is intermediate to that of the parents but also with ongoing gene flow among the parental species, which occasionally results in hybrid formation. We have also found that hybridisation rarely proceeds beyond the $F_1$ stage, as indicated by the low frequency of $F_2$ and backcrossed individuals amongst our sample. This is despite the fact that many *E. paludicola* individuals are mature and produce viable seed, and herbarium collections observed at the State Herbarium of South Australia (AD) that are referred to *E. paludicola* date back to at least the 1920s, suggesting there has been opportunity for multiple generations of interspecific gene flow. Flowering asynchrony might limit the formation of later generation hybrids [39] although both parental species flower over a long period (autumn to spring i.e., April to November) that is coincident with *E. paludicola* (Spring i.e., August to November) [23] suggesting this is also unlikely. Interestingly, the majority of the $F_2$ and all of the unambiguously assigned backcrossed individuals were associated with the revegetation sites that were included in our study. The production of these individuals included seed germination and on-growing seedlings before planting in restoration sites enabling survival of F2 recruits. In contrast, the low frequency of post-$F_1$ hybrids among natural stands suggests low germination and/or survivorship under field conditions. Similar results have been found elsewhere among eucalypts, and for example, Shepherd and Lee [40] note an absence of mature post-$F_1$ interspecific *Corymbia* hybrids in the field, despite the apparent vigour and fecundity of $F_1$ individuals and their cultivated offspring.

Intrinsic outbreeding depression (OBD) provides a plausible explanation for the poor performance of hybrids reported in a number of controlled eucalypt crosses (e.g., [16,17,40–42]) and is manifested by factors including reduced seed viability, delayed germination and increased mortality throughout the life of a cohort. Compared to $F_1$s, recombination and segregation in later generation crosses may result in advanced generation hybrid breakdown due to disruption of coadapted gene complexes, or loss or duplication of chromosomal segments [4]. Minor disadvantages in performance of the hybrids between seed formation and maturity may be expressed by low survivorship and reproductive isolation can be maintained between parent species despite weak barriers to gene flow [43]. A recent study of *E. globulus* × *E. nitens* controlled crosses found that OBD may not be evident in advanced generations until age 2 C4 years and increased with age at least up to 14 years [16]. Late acting postzygotic isolation has also been implicated as a barrier to the formation of later generation hybrids in the field among eucalypts [40] and other long-lived tree species. e.g., [43]. Importantly, the likelihood and intensity of OBD is strongly predicted by evolutionary/taxonomic distance between eucalypt species [11,17]. OBD may provide a reproductive barrier between *E. ovata* and *E. cosmophylla,* which is suggested by:

(1) the globally low numbers of *E. paludicola* individuals, indicating that hybrid formation is infrequent; and (2) the high proportion of $F_1$ individuals within *E. paludicola*, consistent with selection against later generation hybrids.

Among eucalypts, the lack of evidence for hybrid speciation is perhaps surprising given their propensity for hybrid formation [22]. Eucalypts are remarkably constant in their base chromosome number [44–46] and this in part explains why so many natural hybrids occur, and why fertile synthetic interspecific hybrids are relatively easily obtained [45]. An implication of the above is that hybridisation leading to species formation would likely proceed without a change in chromosome number (i.e., homoploid hybrid speciation), a situation that has been considered vanishingly rare [2,47]. Hybrids can suffer reduced fertility and viability and may be intermediate in ecology and are therefore less fit than the parents in the parental habitat. Even in instances where the hybrid is fertile and locally adapted, weak barriers to gene flow between the hybrid and parental species present a substantial barrier to the development of reproductive isolation. The results of our study suggest that while reproductive isolation is incomplete, strong postmating isolation between *E. ovata* and *E. cosmophylla* is manifest by the production of a transient hybrid taxon, and there is no support for the recognition of a stable hybrid species.

*Implications for Conservation*

A hybrid origin for *E. paludicola,* which is currently listed under the EPBC Act as an endangered species, has implications for both its taxonomic and conservation status. Guidelines such as those outlined in the Biodiversity Conservation Act 2016 in the state of Western Australia [48] provide an objective basis to consider the status of *E. paludicola.* Under this scheme, hybrid entities must meet three criteria to warrant conservation listing: they are a distinct entity that is self-perpetuating and has arisen via natural processes. As a set of early generation, predominantly $F_1$ hybrids that have arisen multiple times via rare spontaneous events, we find little support for the ongoing conservation listing of *E. paludicola* or indeed recognition as a distinct species. A better conservation action would be to provide ongoing possibilities for the formation of the hybrid offspring of the two progenitor species, between *E. ovata* and *E. cosmophylla*.

The outcomes of hybridisation are diverse, ranging from the generation of new stable entities through to the loss of diversity through the erosion of species boundaries [1]. While hybridisation may have negative impacts on biodiversity, it has also been argued that hybrids arising from natural processes—and that are not a threat to the integrity of the parental species—should be afforded protection on the basis of preserving genetic diversity and natural evolutionary processes [49,50]. We detected a low level of backcrossing in the *E. paludicola* hybrid system suggesting that hybridisation is unlikely to present a threat to the demographic viability or genetic integrity of either parental species, both of which are relatively common in the region. On the other hand, it has been suggested that hybridisation may have potential adaptive benefits in the context of rapidly changing environments [8] and may have important ecosystem consequences (e.g., [51]). As a relatively rare example of natural intersectional hybridisation within *Eucalyptus, E. paludicola* is of scientific interest, for example, in unravelling the nature of reproductive isolation and speciation within the genus (e.g., [35]). However, it is difficult to justify active and ongoing management for *E. paludicola* beyond the maintenance of the parental species and their habitat. Several conservation listed eucalypt taxa have previously been found to be hybrids in light of molecular evidence (e.g., [6,9]). The results of our study further highlight the limitations of morphological evidence alone in delimiting conservation priorities [48] and in particular for identifying eucalypt hybrids [14] and their systems [17]. It points to the need of thorough molecular genetic analyses as part of any conservation assessment process where hybridisation is likely.

**Supplementary Materials:** The following are available online at http://www.mdpi.com/1424-2818/12/12/468/s1, Table S1: Samples and Genetic Diversity. Table S2: Primers developed for *Eucalyptus paludicola*. Table S3: Input data for the Structure and Newhybs analyses. Table S4: Structure and Newhybs final results. Table S5: NewHybrids simulated data.

**Author Contributions:** Conceptualization, M.W. and D.B. in consultation with the Department for Environment and Water *Eucalyptus paludicola* recovery team; methodology, K.-j.v.D., E.B., D.B., J.Q., H.C.; formal analysis, K.-j.v.D. and E.B.; investigation, E.B., M.W., K.-j.v.D., resources, M.W., D.B., J.Q.; data curation, K.-j.v.D. and M.W.; writing—original draft preparation, M.W., E.B. and K.-j.v.D.; writing—review and editing A.H.T., M.W., E.B. and K.-j.v.D.; visualization, K.-j.v.D.; supervision, M.W.; project administration, M.W., K.-j.v.D.; funding acquisition, M.W. All authors have read and agreed to the published version of the manuscript.

**Funding:** This research received was funded by the South Australian Department for Environment and Water (*Eucalyptus paludicola* recovery program) and The State Herbarium of South Australia.

**Acknowledgments:** Project support provided by Kylie Moritz (Natural Resources Adelaide and Mount Lofty and Natural Resources Murray Darlin Basin, Department for Environment and Water; current address Landscape South Australia—Murraylands and Riverland) for support with collection of samples, liaison with the *Eucalyptus paludicola* recovery team, advice on management actions. Technical support was provided by Ainsley Calladine (State Herbarium of South Australia), Martin O'Leary, samples were provided by Luke Price and Kym Ottewell. We thank the two anonymous reviewers whose comments helped improve the final manuscript.

**Conflicts of Interest:** The authors declare no conflict of interest.

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
