# Peer review of "Genomic Screening Reveals That the Endangered Eucalyptus paludicola (Myrtaceae) Is a Hybrid"

_diversity, doi:10.3390/d12120468_

Round 1
Reviewer 1 Report
I enjoyed this report. I am not an expert in the analytical techniques applied but the methods and data are presented clearly enough for a non-specialist to follow.
The background of the study group is clearly presented, as is the context in which the questions are asked. The identification of hybrids is useful not only for questions of conservation but also for systematics and derivative studies so this is a neat review and case study in the Eucalypts.
Methods are logical, appear appropriate and are clearly presented. The piece is well written and there are no issues with the English.
Discussion is clear and easy to follow and provides a useful summary of possible causes and consequences of different types of hybridisation. Although it might seem obvious, I had never heard the term 'transient hybrid' so I suggest including a reference to the term or a brief definition or discussion of stable vs transient hybrids.
In my opinion this represents a valuable contribution and should be accepted in its present form.
Author Response
We thank reviewer one for their extremely quick and positive review. The one comment that we could address is as follows:
Reviewer 1 comment: I had never heard the term 'transient hybrid' so I suggest including a reference to the term or a brief definition or discussion of stable vs transient hybrids.
Reply: We have added a definition for transient hybrids on line 82 where we first mention the term.
Reviewer 2 Report
The paper is sound and well-written and the methods adequate. I have only minor comments.
1) There are many populations in the study, but the Structure is run with k= 1 to 8 only. Authors should explain the reasons.
2) The number of individuals per population varies widely, and there are many populations of the hybrid compared to the populations of the parentals. These factors, especially the relative number of populations, may cause a strong bias in the results of genetic diversity if you compare genetic diversity of hybrid and parentals. Authors should be extremely prudent in the discussion on this point.
3) Authors conclude that E. paludicola is a transient F1 hybrid that does not deserve particular conservation measures. Transiency of the hybrid is, in my view, not completely proven, because it is isolated from the parentals (no backcrossing) and there are at least some F2. I would tune down this concluding remark, because it would have very negative impact on the nothospecies if it is finally found to be stable and intersectionals hybrids in Eucaliptus are rare.
4) Figure 2, acronym BC (backcrossed?) should be explained in the figure caption.
5) Table S2 is mentioned in the text, but is not listed in the caption of Suppplementary Material. Viceversa, Tables S3 and S5 are listed in the caption, but they are not mentioned in the text.
Author Response
We thank reviewer two for their quick and thorough review. We have addressed their main concerns with the manuscript and outline our replies point by point below:
Reviewer 2 comment: There are many populations in the study, but the Structure is run with k= 1 to 8 only. Authors should explain the reasons.
Reply: We were interested in the highest level of structure in our data (i.e. how many species). Given that we expected at most, to find 3 species, we believe that K=8 adequately accounts for our sampling.
Reviewer 2 comment: The number of individuals per population varies widely, and there are many populations of the hybrid compared to the populations of the parentals. These factors, especially the relative number of populations, may cause a strong bias in the results of genetic diversity if you compare genetic diversity of hybrid and parentals. Authors should be extremely prudent in the discussion on this point.
Reply: The parents come from two different Eucalyptus sections. Given that fact, we think that sampling more parents would not change the results because the STRUCTURE results are already quite strong. Smaller sampling would be more problematic if the three taxa were sister to each other. We are aware of these issues with Structure, but the same inferences were made using different methods with different underlying assumptions (i.e. PcOA). We believe that this is a consequence of the high level of genetic differentiation between parental populations leading to strong genetic structuring.
Reviewer 2 comment: Authors conclude that E. paludicola is a transient F1 hybrid that does not deserve particular conservation measures. Transiency of the hybrid is, in my view, not completely proven, because it is isolated from the parentals (no backcrossing) and there are at least some F2. I would tune down this concluding remark, because it would have very negative impact on the nothospecies if it is finally found to be stable and intersectionals hybrids in Eucalyptus are rare.
Reply: The absence of backcrossing would be consistent with reproductive isolation in the case where the E. paludicola individuals form a distinct genetic cluster. Rather, we found that E. paludicola is intermediate between the parents, and the absence of backcrossing is most likely a consequence of hybrid inviability. This is further supported by the virtual absence of F2 individuals, other than those sampled from revegetation sites (i.e. nursery raised). While it is difficult to ‘completely prove’ this point, our data are entirely consistent with a ‘transient hybrid’ hypothesis and we feel sufficient to justify the conservation recommendations we have made.
Reviewer 2 comment: Figure 2, acronym BC (backcrossed?) should be explained in the figure caption.
Reply: We have added to the caption that BC stands for ‘back cross’.
Reviewer 2 comment: Table S2 is mentioned in the text, but is not listed in the caption of Supplementary Material. Viceversa, Tables S3 and S5 are listed in the caption, but they are not mentioned in the text.
Reply: We think we have confused the reviewer with how we have labelled our figures and tables in the supplementary materials. We have attempted to We have five supplementary files. Some of these files are excel spreadsheets that have some more than one figure embedded in them. This is why some references said Supp. 2 Fig S2. We have rearranged the text to say for instance ‘Fig S2 in Supp.2’ to make it clearer in the instances where a figure is embedded in a supplementary file.